# Health Literacy, Vaccine Confidence and Influenza Vaccination Uptake among Nursing Home Staff: A Cross-Sectional Study Conducted in Tuscany

**DOI:** 10.3390/vaccines8020154

**Published:** 2020-03-30

**Authors:** Chiara Lorini, Francesca Collini, Francesca Gasparini, Diana Paolini, Maddalena Grazzini, Francesca Ierardi, Giacomo Galletti, Patrizio Zanobini, Fabrizio Gemmi, Guglielmo Bonaccorsi

**Affiliations:** 1Department of Health Science, University of Florence, 50134 Florence, Italy; francesca.gasparini1@stud.unifi.it (F.G.); diana.paolini@unifi.it (D.P.); patrizio.zanobini@unifi.it (P.Z.); 2Quality and Equity Unit, Regional Health Agency of Tuscany, 50141 Florence, Italy; francesca.collini@ars.toscana.it (F.C.); francesca.ierardi@ars.toscana.it (F.I.); giacomo.galletti@ars.toscana.it (G.G.); fabrizio.gemmi@ars.toscana.it (F.G.); 3Health Management Unit, Careggi Teaching Hospitals, 50134 Florence, Italy; grazzinim@aou-careggi.toscana.it

**Keywords:** influenza vaccination, nursing homes, health literacy, vaccine confidence, staff

## Abstract

The aim of this cross-sectional study is to address whether health literacy (HL) and vaccine confidence are related with influenza vaccination uptake among staff of nursing homes (NHs). It was conducted in Tuscany (Italy) in autumn 2018, including the staff of 28 NHs. A questionnaire was used to collect individual data regarding influenza vaccination in 2016–2017 and 2017–2018 seasons; the intention to be vaccinated in 2018–2019; as well as demographic, educational, and health information. It included also the Italian Medical Term Recognition (IMETER) test to measure HL and eight Likert-type statements to calculate a Vaccine Confidence Index (VCI). The number of employees that fulfilled the questionnaire was 710. The percentage of influenza vaccination uptake was low: only 9.6% got vaccinated in 2016–2017 and 2017–2018 and intended to vaccinate in 2018–2019. The VCI score and the IMETER-adjusted scores were weakly correlated (Rho = 0.156). At the multinomial logistic regression analysis, the VCI was a positive predictor of vaccination uptake. In conclusion, vaccine confidence is the strongest predictor of influenza vaccination uptake among the staff of NHs. The development of an adequate vaccine literacy measurement tool could be useful to understand whether skills could be related to vaccine confidence.

## 1. Introduction

Influenza is a highly contagious viral infection with global circulation: The World Health Organization (WHO) estimates that influenza annually infects about 5%–15% of the population worldwide. Illnesses range from mild to severe and even death: hospitalization and death occur mainly among high risk groups [1]. One of the most vulnerable groups for severe disease and influenza-related complications is elderly people: globally, 67% of the deaths have occurred among people 65 years and older with large regional variation, from 36% in Sub-Saharan Africa to 86% in Europe [2]. In particular, in this age class, those living in nursing homes (NHs) suffer from higher likelihood of infection, getting sick and having comorbidities that increase vulnerability to poor outcome following infection [3,4,5,6].

Influenza vaccination is effective in reducing the burden of influenza illness among elderly people in NHs: recent studies conducted in this setting demonstrated the role of influenza vaccination in predicting elderly survival [7,8] and in preventing deaths and hospitalization due to influenza disease [9]. However, elderly people may be insufficiently protected by vaccination due to the immunosenescence that accompanies aging [10]. For this reason, in order to reduce influenza exposure for these older adults with frailty, infection control policies and procedures must be implemented and primarily vaccination of the NHs staff that give assistance to them in daily living is required [11]. This is particularly relevant considering that, as for what has been observed for healthcare workers in other settings (i.e., hospitals), many studies described low levels of annual influenza vaccination, reporting multiple concerns that lead to hesitancy to receive influenza vaccination [12,13,14]. Specifically, lack of knowledge on the risk related to influenza and concern regarding the vaccine have frequently emerged as determinants of vaccine hesitancy among NH staff while vaccinated employees reported to get the vaccine to protect themselves and the elderly they care for [11,12,13,14].

Vaccine confidence is defined as the trust in the effectiveness and safety of vaccines, as well as the trust in the healthcare system that delivers them; high confidence in vaccination programmes, together with low complacency and high convenience of vaccine, are crucial for maintaining high coverage rates [15]. From recent findings it emerged that higher vaccine confidence among healthcare workers could result in a larger proportion of the general population expressing positive vaccination beliefs, of vaccine acceptance by either the general population or the target groups, as well as of vaccine acceptance by themselves, although with differences by geographical areas and vaccine [16,17,18,19]. Recently, a Vaccine Confidence Index (VCI) has been developed in order to objectively measure the vaccine confidence, with findings that indicate the viability of this approach to measure vaccine-related confidence (that is, sentiments that influence vaccination behaviours), and, more broadly, to illustrate the relationships between these sentiments and public attitudes towards health services [19,20]. A modified version of the VCI has been recently applied in a study conducted in a sample of healthcare workers, highlighting the usefulness and the versatility of such an index in understanding determinants of vaccine hesitancy and vaccine acceptance [16]. To the best of our knowledge, to date a similar approach has not been applied in the NHs setting.

Health literacy (HL) deals with the capacities of people to meet the complex demands of health in a modern society [21]. The Sørensen Integrated Model states: “health literacy is linked to literacy and entails people’s knowledge, motivation, and competence to access, understand, appraise, and apply health information in order to make judgments and take decisions in everyday life concerning healthcare, disease prevention, and health promotion to maintain or improve quality of life during the life course” [22]. HL is therefore the balance between individual, community and population skills and the system’s complexity: people’s abilities are mediated by environmental demands and situational complexities, i.e., the context in which HL is developed and applied [23,24]. As for HL, vaccine literacy has been defined as “not simply knowledge about vaccines, but also developing a system with decreased complexity to communicate and offer vaccines as *sine qua non* of a functioning health system” [25]. From a theoretical point of view, health and vaccine literacy can be considered as determinants of vaccine confidence [26,27]. Nonetheless, to date, studies investigating the relationship between health or vaccine literacy and vaccination uptake are scarce and have presented inconsistent results: the association varies by population groups, vaccines, geographical areas, and measures of health or vaccine literacy applied [26,27]. Furthermore, to the best of our knowledge, no studies have already demonstrated the association between health or vaccine literacy and vaccine confidence in different healthcare settings. Moreover, to date no measurement tools for vaccine literacy targeted to adults working in a healthcare setting have been developed.

The aim of this study is to address whether HL and vaccine confidence affect influenza vaccination uptake among staff of NHs. The research queries are the following:Does HL influence vaccination uptake among staff of NHs?Does vaccine confidence influence vaccination uptake among staff of NHs?Are HL and vaccine confidence related?

## 2. Materials and Methods

The study adopted a cross-sectional design and was conducted according to the principles of the Helsinki Declaration. It was proposed to the Chief Officers of each Tuscan NHs (about 300) and 28 subjects voluntarily joined. In each NH, all the employed staff members were included, regardless of the type of employment contract, the job responsibilities and the qualification.

### 2.1. Questionnaires

Data was collected using two questionnaires. The first one was fulfilled by each Chief Officer to collect general information on the NH (number of hosted residents, mean time of stay of the residents, percentage of residents vaccinated against influenza in the 2017–2018 season, number of employees, ownership type, policy, and practices for influenza vaccination of staff). The second one was fulfilled by the staff members to collect individual data regarding influenza vaccination (self-reported) in 2016–2017, in 2017–2018 or intention to be vaccinated in 2018–2019 seasons; knowledge, awareness and attitudes concerning influenza and influenza vaccination; as well as demographic, educational, and health information. As far as health information is concerned, data on chronic cardiovascular, renal, respiratory, and autoimmune diseases were collected, as well as an assessment of self-perceived health status (from “1”—bad— to “10”—excellent). This questionnaire was developed adapting the one used in a previous study conducted in Tuscan hospitals [28,29,30]; the questionnaire included also the Italian Medical Term Recognition (IMETER) test to measure HL [31,32]. The staff questionnaire had no individual identifiers to encourage completion but had a NH identifier. Both questionnaires were self-administered either in paper and pencil or computer-based form, as preferred by participants. The survey was conducted in September–October 2018.

### 2.2. Health Literacy Measure: The IMETER Test

As the original test that was developed in English language [33], the IMETER is an objective measure of functional HL, i.e., one dimension of the concept of HL that entails the basic skills in reading and writing that are necessary to function effectively in everyday situations related to health [34]. It is based on word/non-word recognition: It is composed of a list of 70 terms (40 real medical and 30 non-real medical words, that intuitively sound like real medical terms) and the interviewed persons were asked to check-off those they recognized as actual medical words. HL levels were defined as the number of words correctly recognized, with higher scores reflecting higher HL. The “adjusted-score” was calculated as the number of words correctly checked-off, minus the number of non-actual words checked-off. Since the adjusted-score proved to better predict health behaviour than the unadjusted-score [32], this was chosen. According to the final score, the HL level was classified as follows: 0–20 = low (cutoff score), 21–34 = marginal, 35–40 = functional HL [33].

### 2.3. Calculation of the Vaccine Confidence Index

A Vaccine Confidence Index (VCI) was calculated according to the literature [16,19,20], considering eight Likert-type statements included in the staff questionnaire to which the participants were asked to declare their agreement or disagreement. The statements were the following:Influenza is a serious illness (A1)Influenza vaccine is effective (A2)Healthcare workers must get vaccinated (A3)By getting vaccinated I protect people close to me from influenza (A4)it is better to contract influenza than to get the vaccination (B1)Influenza vaccines have serious side effects (B2)Vaccine can cause influenza (B3)Opposed to vaccination (B4)

The level of agreement or disagreement was scored as follows: “totally agree” = 1; “partially agree” = 2; “partially disagree” = 3; “totally disagree” = 4. For the first four statements (A1–A4), the higher the Likert score, the better the propensity towards vaccines while for the second four (B1–B4), the higher the Likert score, the lower the propensity.

The vaccine confidence index was calculated as follows:VCI = [ (A1+A2+A3+A4)/4]/[(B1+B2+B3+B4)/4].
were A1, A2, A3, and A4 were the scores to the first four statements while B1, B2, B3, and B4 were those of the second four.

### 2.4. Statistical Analyses

Data was presented as percentage or as mean, standard deviation and quartiles. Numerical variables were tested for normality using the Kolmogorov–Smirnov test.

Information regarding self-reported influenza vaccination uptake and intention to vaccinate were grouped as follows, according to the consistency of the influenza vaccination during the years: “vaccinated in 2016–2017, 2017–2018 and intention to vaccinate in 2018–2019” (always get vaccinated); “vaccinated at least one time in 2016–2017, in 2017–2018 or intention to vaccinate in 2018–2019” (sometimes get vaccinated); “never vaccinated in 2016–2017, in 2017–2018 and not intentioned to get the vaccine” (never get vaccinated).

Association between the collected data and vaccination were assessed using Chi^2^, Student t test, ANOVA, or the corresponding non-parametric tests. Pearson or Spearman correlation analysis was performed including the score at the IMETER test and the VCI score, by job qualification and reported vaccination uptake.

A multinomial (polytomous) logistic regression model was applied in order to quantify the role of HL and VCI in predicting the reported vaccination uptake or the intention to get the vaccine. This model is used for a categorical dependent variable with outcomes that have no natural ordering and generalized the binary logistic regression to a nominal dependent variable with more than two categories. In particular, the dependent variable (outcome) is the reported influenza vaccination uptake and intention to vaccinate (as previously described: “never get vaccinated”; “sometimes get vaccinated”; “always get vaccinated”), with “never get vaccinated” as the reference. The independent variables (covariates) were sex, age, mother language, the IMETER adjusted-score, the VCI value, and the variables significantly associated with the outcome variable at the univariate analysis (living with elderly people, suffering from autoimmune diseases or from renal chronic diseases). The regression coefficients were expressed as relative risk ratio (RRR).

All statistical tests were two-sided, and p-values were considered statistically significant when below 0.05.

The analyses were conducted using Stata software version 14 (StataCorp LP, College Station, TX, USA).

## 3. Results

### 3.1. Characteristics of the Nursing Homes

Table 1 summarizes the characteristics of the NHs included in the study. The NHs showed a high variability in the investigated variables, including the response rate of the staff to the survey. None of the Chief Officers reported having a medical doctor as a permanent member of the staff while three of them reported to receive periodic advices and visits by a geriatrician. As the latter figures have only sporadic contact with the residents, they have been excluded from the survey. It is important to note that, in Italy, medical assistance in NHs is guaranteed by the general practitioner of each resident.

### 3.2. Characteristics of the Staff

The number of employees that fulfilled the questionnaire was 710 and the collected data are described in Table 2 and Table 3. They were mainly females (80.1%), with high school degree (43.3%), assistant/aide (51.3%), in good health (about 74% reported a score equal or higher than 7 in self-perceived health status), and Italian was the mother tongue (82.4%). The mean age was 43.3 years (range: 20–70). About 40% declared to live with people belonging to a high-risk group and 15% to suffer from a chronic condition considering at risk for influenza. Regarding the results of the IMETER test, 27.3% had low HL, 60.6% marginal HL and 12.1% functional HL. The median score at the VCI was 1.5 (range: 0.3–4).

The percentage of reported influenza vaccination uptake in the 2016–2017 season was 16%, those in 2017–2018 was 16.6%, and regarding the intention to get the vaccine in 2018–2019, this was 28.4%.

Combining the information regarding reported influenza vaccination uptake, 9.6% always get vaccinated, 28% sometimes get vaccinated, 62.1% never get vaccinated, and 0.3% (N = 2) did not declare their vaccination status and the intention to get vaccinated.

Age, IMETER adjusted-score, VCI score, and self-perceived health score were not normally distributed.

### 3.3. IMETER, VCI and Influenza Vaccination

The percentage of always gets vaccinated was significantly higher for those who declared to live with elderly people (14.7% always get vaccinated), or at least with one subject with the listed characteristic (less than 9 years old, elderly people, or people with chronic diseases: 11.8% always get vaccinated), and for those suffering from renal chronic disease (60% always get vaccinated) (Table 2). Those suffering for autoimmune disease more frequently declared to sometimes get the vaccine (44.2%) (Table 2). Moreover, the VCI was significantly higher with the increase of getting vaccinated (Table 3). Nonetheless, among those who reported a high level of VCI score (higher than 75°centile, equal to 2.3), 62 (36.4%) never get vaccinated and among those who reported low level of VCI score (lower than 25°centile, equal to 1.1) only one (0.6%) always get vaccinated and 25 (15.1%) sometimes get vaccinated.

The IMETER adjusted-score and the VCI were significantly, although not strongly, correlated (Rho = 0.156; *p* < 0.001). This correlation was not statistically significant among those who always get vaccinated (Rho = 0.166; *p* = 0.189), while it was significantly associated both among who sometimes gets vaccinated (Rho = 0.304; p < 0.001) and who never gets vaccinated (Rho = 0.137; *p* = 0.005) (Figure 1).

Table 4 reports the results of the multinomial logistic regression analysis. After having adjusted for sex, age and mother tongue, VCI maintained its significant role in predicting the outcome variable, both among who sometimes gets the vaccine and who always gets the vaccine. Specifically, considering “never get vaccinated” as the reference category, the likelihood of “sometimes get vaccinated” is 180% higher with the increase of 1 point in the VCI score (RRR = 2.8) while those “always get vaccinated” is 469% higher (RRR = 5.69). On the other hand, the IMETER adjusted-score was not significantly associated with the reported vaccination uptake. Living with elderly people maintained its positive predictive role in predicting who “always gets vaccinated” (RRR = 3.46) as well as suffering from renal chronic diseases (RRR = 56.6), although the latter presented a very large 95% confidence interval, probably due to the low number of respondents with this condition (N = 5). Moreover, suffering from autoimmune diseases confirmed its positive, predictive role of “sometimes get vaccinated” (RRR = 2.56).

## 4. Discussion

The aim of this cross-sectional study was to address whether HL and vaccine confidence are related with influenza vaccination uptake (self-reported) among the staff of 28 Tuscan NHs, including both healthcare and non-healthcare workers. No previous studies have been conducted by now including both a measure of HL and one of vaccine confidence in similar target groups, so that comparisons are limited.

### 4.1. Influenza Vaccination

First of all, the results showed that influenza vaccination is uncommon among the staff of NHs: the percentage of influenza vaccination uptake (self-reported) in each investigated season was about 16%, while the intention to get the vaccine in 2018–2019 was 28.4%. Combining the data of the three seasons, 9.6% gets vaccinated in 2016–2017, 2017–2018 and intended to vaccinate in 2018–2019; 28% gets vaccinated in at least one of the two seasons or expressed the intention to get the vaccine in 2018–2019; while 62.1% neither gets vaccinated nor expressed the intention to get the vaccine. This data describes a worse scenario than that presented in the 2016–2017 season for the USA, where the 68% of the staff of long-term care facilities got influenza vaccine [35], and that observed in Tuscany in a sample of paid non-familial caregivers of elderly people assisted at home, where 36.2% got it [36]. On the other hand, the results are similar to those described in the same geographical area (Tuscany) among hospital healthcare workers [28].

### 4.2. Health Literacy, Vaccine Confidence and Influenza Vaccination

For what concerns HL, to the best of our knowledge, no previous studies have investigated this aspect among the staff of NHs, as in other healthcare settings: researchers’ attention has been devoted by now primarily to assess the HL of the patients or of the organization; the lack of data could be due to the assumption that having a specific education in healthcare matters, as a degree in nursing or whatsoever, or having worked in healthcare settings could be sufficient for developing high levels of HL. In fact, university students of medical or biological courses presented higher functional HL than those attending non-scientific courses [31]; on the other hand, in a population-based study conducted in Florence (Italy), no significant differences in functional HL have been described between people with working experience in healthcare settings and those without [37]. Moreover, in the same geographical area as well as in other countries, paid non-familial caregivers of the elderly with disabilities assisted at home presented low levels of functional HL [38,39]. Finally, it is important to note that the staff of NHs is composed of different professionals with different levels and types of education, involved in many daily activities of the residents; specifically, in our sample, no medical doctors were included since they were not part of the staff and most of the respondents were assistants/aides. In fact, in our study, only 12.1% of the respondents presented the highest level of HL; furthermore, mean and median values of IMETER adjusted-score (respectively 23.4 and 28) showed a lower level of HL than those described in other settings (adults, university students) using the same HL measure [31,32].

The results of this study showed no significant association between HL and self-reported influenza vaccination uptake, as also described in another research conducted on paid non-familial caregivers of elderly with disabilities assisted at home, in the adult population as well as in pregnant women [26,36,40], while for the elderly population, a positive predictive role of HL in influenza vaccination uptake has been described [26]. Differently from the results of this study, other preventive behaviors, such as cancer screening or the intake of fruits and vegetables, have been positively associated with HL by many authors, with differences related to the study population or the measurement tool of HL that was used [41,42,43].

Measuring vaccine confidence is an emerging science [20]. The VCI value reported in this study is similar to that reported for nurses and assistant/aides in a study conducted in a Tuscan hospital [16]; in that study, in which HL was not assessed, the results of a multivariate regression analysis showed that the VCI score was a significant predictor of vaccination uptake, with a degree of association similar to that calculated in our study. For this reason, the VCI is confirmed to be useful in analyzing the drivers of vaccination uptake and monitoring vaccine confidence over time, as also suggested by Larson et al. [20]. Nonetheless, since the results of this study reported that in some cases high confidence in vaccine had not led to vaccination uptake and vice versa, future investigation will be useful to understand the weight of the other drivers of influenza vaccination acceptance/refuse.

Moreover, data suggests that HL, measured using the IMETER, and vaccine confidence, measured using a VCI, are quite dissimilar: general HL competences, particularly those related to basic abilities to understand words in a medical setting, are weakly related to confidence in vaccine. This correlation seems to be stronger among those people who do not get the influenza vaccination every season, while among those who always or never get the vaccine this correlation is lower. The development of a vaccine literacy measure for adults, including not only functional but also other dimensions of literacy, and its application in studies similar to this one could contribute to better investigate the role of vaccine skills in predicting confidence and acceptance. In particular, the measures of interactive (more advanced cognitive and literacy skills which, together with social skills, can be used to actively participate in everyday situations, extract information and derive meaning from different forms of communication, and apply this to changing circumstance) and critical skills (more advanced cognitive skills which, together with social skills, can be applied to critically analyze information and use this to exert greater control over life events and situations) [34] related to vaccine could help to quantify the level of autonomy and personal empowerment in vaccine-related decision-making, that could be more related to trust in the effectiveness and safety of vaccines, as well as in the healthcare system that delivers them. Moreover, interactive and critical HL skills could be more related to the emotional aspects that affect vaccine confidence. In fact, emotions such as anger, fear, and regret, as well as medical mistrust, may impact on vaccine confidence, motivation to accept vaccination and vaccination uptake. It is important to note that, according to the Sørensen Integrated Model, motivation is included in the definition of HL.

### 4.3. Limitations of the Study

This study has some limitations. First, it has been conducted using a convenience sample: the staff was recruited within those NHs whose Chief Officer voluntarily joined the study (about 10% of all Tuscan NHs). This aspect could introduce a selection bias related to the interest of the directors in participating in a study on this topic. Moreover, in some NHs, the percentage of the staff that fulfilled the questionnaire was quite low; since it was not possible to investigate the reasons for low compliance in some facilities, this aspect could have introduced another selection bias. Overall, these aspects lead to limitations in the generalizability of the results. Another limitation of the study is related to the measure of both HL and vaccine confidence. As far as HL is concerned, many measurement tools have been developed by now, exploring the many dimensions and domains of such a complex concept in different ways [44]. The one used in this study has been developed to measure functional HL. Other studies have reported that the use of measures of general HL for predicting vaccine uptake could undergo misinterpretation of the relationship between skills and vaccination acceptance [26,27]. Moreover, the VCI score was calculated according to what has been already described but not using precisely the same questions [16,19,20].

## 5. Conclusions

According to the results of the study, vaccine confidence is the strongest predictor of self-reported influenza vaccination uptake among the staff of NHs. Future studies will have to be performed in order to assess the determinants of vaccine confidence in such a target group, in order to identify the strategies to improve influenza vaccination uptake. The development of an adequate vaccine literacy measurement tool could be useful to understand whether specific functional, critical and interactive skills could influence vaccine confidence.

## Figures and Tables

**Figure 1 vaccines-08-00154-f001:**
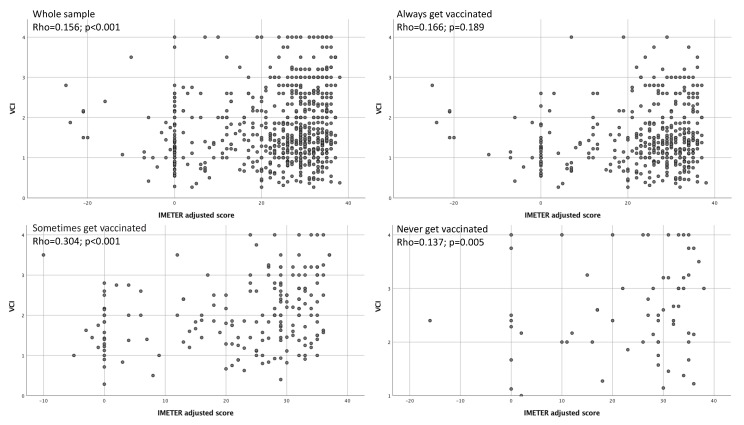
IMETER adjusted-score and Vaccine Confidence Index (VCI): Spearman correlation analysis in the whole sample and by vaccination.

**Table 1 vaccines-08-00154-t001:** Characteristics of the nursing homes (N = 28).

Variables	Range	Centiles		
		25°	50°	75°
Number of residents	17–118	32	43	59
Staff (total)	22–146	32.7	50	62.7
Staff: number of				
Medical doctors *	0–1	0	0	0
Nurses	3–14	4	5.5	7.5
Physiotherapists	1–4	1	2	2.7
Assistants/aides	11–91	19.2	22.5	29.7
Health educators	1–5	1	2	2
Cleaning staff	1–14	2.2	3.5	6.7
Other nonclinical staff	0–19	1.2	6	7
Percentage of vaccinated residents against influenza	22.5–100	79	89	97
Percentage of respondents among staff members	15.2–92.1	30.1	51.5	67.4
	**N**	**%**		
Mean time of stay of the residents				
12 months or less	2	7.1		
More than 12 months	26	92.9		
Ownership type				
Public	14	50		
Private for-profit	5	17.9		
Private not-for-profit	9	32.1		

* Three NHs reported “1” as the number of medical doctors.

**Table 2 vaccines-08-00154-t002:** Characteristics (categorical variables) of the staff (N = 711): total sample and reported influenza vaccination uptake in 2016–2017, 2017–2018 seasons and intention to vaccinate in 2018–2019.

Variables	Total Sample		Reported Influenza Vaccination Uptake (Row Percentage) *			
	N	%^#^	Always Gets Vaccinated N = 68; 9.6%)	Sometimes Gets Vaccinated (N = 199; 28%)	Never Gets Vaccinated (N = 441; 62.1%)	*P*-Value (Chi^2^ Test)
**Sex**						
Males	106	14.9	11.3	32.1	56.6	0.406
Females	569	80.1	9	27.6	63.4	
**Mother Language**						
Italian	585	82.4	9.6	27.1	63.4	0.329
Others	64	9.0	4.7	32.8	62.5	
**Educational level**						
Less than high school diploma	180	25.4	11.1	26.1	62.8	0.884
High school degree	308	43.3	8.4	28.6	63	
Bachelor’s degree and higher	183	25.8	9.8	26.8	63.4	
**Qualification**						
Nurses	93	13.1	12.9	32.3	54.8	0.407
Physiotherapists	37	5.2	8.1	18.9	73	
Assistants/aides	364	51.3	7.7	29.5	62.8	
Health educators	25	3.5	20	16	64	
Other clinical staff	40	5.6	7.5	27.5	65	
Cleaning staff	45	6.3	8.9	31.1	60	
Other nonclinical staff	60	8.5	15	25	60	
**Living with °**						
Children of less than 9 years	142	20	8.5	28.9	62.7	0.742
Elderly people	137	19.3	14.7	31.6	53.7	0.014
People with chronic diseases	74	10.4	12.2	35.1	52.7	0.109
At least one of the previously listed condition	288	40.6	11.8	32.2	56.1	0.017
**Suffering from °**						
Cardiovascular chronic diseases	9	1.3	0	33.3	66.7	0.581
Respiratory chronic diseases	53	7.5	15.1	30.2	54.7	0.294
Renal chronic diseases	5	0.7	60	0	40	0.001
Diabetes	13	1.8	7.7	15.4	76.9	0.599
Autoimmune diseases	43	6	9.3	44.2	46.5	0.028
At least one of the previously listed disease	106	14.9	11.3	34	54.7	0.142
**Health literacy level (IMETER adjusted-score)**						
Low HL	194	27.3	11.9	31.6	56.5	0.176
Marginal HL	430	60.6	34	26.6	65.5	
Functional HL	86	12.1	11	27.9	59.3	
**Reported influenza vaccine uptake**						
2016–2017	114	16	-	-	-	-
2017–2018	118	16.6	-	-	-	
Intention to vaccine uptake in 2018–2019	202	28.4	-	-	-	
Vaccine uptake in 2016–2017, 2017–2018 and intention to vaccinate in 2018–2019	68	9.6	-	-	-	

**^#^** for “sex”, “mother tongue”, “educational level”, and “qualification”, the sum of the percent value by categories is lower than 100 % due to missing. ° For each condition, the missing value varied from 11% to 18%. * Two missing. IMETER= Italian Medical Term Recognition.

**Table 3 vaccines-08-00154-t003:** Characteristics (numerical variables) of the staff (N = 711): total sample and by reported influenza vaccination uptake in 2016–2017, 2017–2018 seasons and intention to vaccinate in 2018–2019.

Variables	Total Sample			Influenza Vaccination			
	Mean ± SD	Median	Range	Always Get Vaccinated (N = 68; 9.6%) *Mean ± SD; Median*	Sometimes Get Vaccinated (N = 199; 28%) *Mean ± SD; Median*	Never Get Vaccinated (N = 441; 62.1%) *Mean ± SD; Median*	*P* Value (Kruskal Wallis Test)
**Age (years)**	43.3 ± 11	44	[20; 70]	44.6 ± 12.6; 47	43.4 ± 11.4; 45.5	43 ± 10.6; 43	0.473
**Self-perceived health status (score)**	8.1 ± 1.6	8	[1; 10]	8 ± 1.6; 8	8.2 ± 1.5; 8	8.14 ± 1.6; 8	0.825
**IMETER adjusted-score**	23.4 ± 12.3	28	[−25; 38]	23.2 ± 12.9; 29	22.8 ± 12.1; 28	23.7 ± 12.4; 28	0.649
**VCI score**	1.7 ± 0.9	1.5	[0.3; 4]	2.6 ± 0.9; 2.5	2 ± 0.8; 2	1.5 ± 0.7; 1.3	<0.001

IMETER = Italian Medical Term Recognition; VCI = Vaccine Confidence Index.

**Table 4 vaccines-08-00154-t004:** Multinomial logistic regression model. Dependent variable: reported influenza vaccination uptake and intention to vaccinate (three levels: “never get vaccinated”, “sometimes get vaccinated”, “always get vaccinated). The reference category of the dependent variable was “never get vaccinated”. Relative risk ratio (RRR) values were adjusted by sex, age, and mother language.

Reported Influenza Vaccination Uptake *	Variables	RRR	P > z	[95% Confidence Interval]
Never (reference)	-	1	-	-
Sometimes	Suffering from renal chronic diseases	4.40 × 10^−7^	0.996	0
	Suffering from autoimmune diseases	2.56	0.043	[1.03; 6.36]
	Living with elderly people	1.37	0.304	[0.75; 2.48]
	IMETER adjusted-score	0.99	0.195	[0.97; 1.01]
	VCI	2.84	0.000	[2.10; 3.84]
	Cons	0.10	<0.001	[0.03; 0.36]
Always	Suffering from renal chronic diseases	56.6	0.003	[4.13; 774.53]
	Suffering from autoimmune diseases	0.95	0.952	[0.18; 5.02]
	Living with elderly people	3.46	0.003	[1.53; 7.83]
	IMETER adjusted-score	0.98	0.315	[0.95; 1.01]
	VCI	5.69	0.000	[3.58; 9.05]
	Cons	0.003	<0.001	[0.001; 0.026]

* Reported influenza vaccination uptake in 2016–2017, 2017–2018 seasons and intention to vaccinate in 2018–2019. “*Cons*” estimates baseline relative risk for each outcome. IMETER = Italian Medical Term Recognition; VCI = Vaccine Confidence Index. N = 441; LR chi2(16) = 121.83; Log likelihood = −319.29; PseudoR2 = 0.16.

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
