# Peer review of "Health Literacy, Vaccine Confidence and Influenza Vaccination Uptake among Nursing Home Staff: A Cross-Sectional Study Conducted in Tuscany"

_vaccines, 2020, doi:10.3390/vaccines8020154_

Round 1

Reviewer 1 Report

The study is well-researched with a large number of staff responding to the questionnaire. The sample is likely to be representative. The conclusions stand alone - but with the extra urgency surrounding coronavirus, the results take on an added importance. Not only should staff set the example by being vaccinated against influenza, they should set the example when it comes to protection against coronavirus.

The sub-grouping and the graphical representation of the data are particularly useful and will help readers understand the significance of the work.

Author Response

The study is well-researched with a large number of staff responding to the questionnaire. The sample is likely to be representative. The conclusions stand alone - but with the extra urgency surrounding coronavirus, the results take on an added importance. Not only should staff set the example by being vaccinated against influenza, they should set the example when it comes to protection against coronavirus.

The sub-grouping and the graphical representation of the data are particularly useful and will help readers understand the significance of the work.

Replay. Thank you very much for the comments. In fact, we are planning a survey to evaluate, among the same sample, any changes in the risk perception of influenza and influenza vaccination during the coronavirus pandemic.

Reviewer 2 Report

The study presents data on the association between confidence in vaccination, health literacy and reported influenza uptake among elderly care workers. The study presents important findings on vaccination uptake.

  1. Abstract: Refrain from words of causality (i.e., “affected”) in a cross-sectional study
  2. Abstract: “slowly correlated” – non-English; did the authors mean “weakly associated”?
  3. Introduction: “complacency .. of vaccination” – unclear.
  4. There is no explanation to the following sentence: “From a theoretical point of view, health and vaccine literacy can be consideredas determinants of vaccine confidence [26,27]. “  - it is definitely not a determinant, as all the variables are measured in a non-experimental design. The authors in their systematic review in 2018 (ref #26) concluded that the relationships “remain unclear”. How has it become a determinant? Which theory? Unfounded, unfortunately.
  5. Methods, under data analysis. There is no information on “uptake” but rather on “reported uptake”. It’s all self-reported. Change that.
  6. Methods, under data analysis, regression. It would make sense to divide the variables to categories and into 3 steps: demo, background (e.g.,having diseases), and perceptual. Not just adjustment for age, sex and language (proxy for immigration?).
  7. I do not think that Figure 1 contributes to the manuscript. Why is it presented? There is no “story” in it. Authors may consider removing it.
  8. Under discussion. Refrain from using the words “affect” “influence”. Inappropriate in a cross-sectional design.
  9. Under discussion. Pls discuss the findings on the association between HL and vaccination in the context of findings on the association between HL and health behaviors. See the reviews by Berman (2011) and Neter & Brainin (2019).
  10. Under discussion. “since the results of this study reported that in some cases
    high confidence in vaccine had not led to vaccination uptake and vice versa” – unclear. Do the authors refer to the fact that there is distribution? To the association bot being 1?
  11. The authors explain the lack of association between HL and confidence in vaccination only in terms of cognitive and behavioral (skill) variables. Pls consider also emotions.
  12. /typo: “people.. does” into “people .. do”

Author Response

The study presents data on the association between confidence in vaccination, health literacy and reported influenza uptake among elderly care workers. The study presents important findings on vaccination uptake.

Abstract: Refrain from words of causality (i.e., “affected”) in a cross-sectional study

Reply. The term “affected” has been changed (line 18), as well as the term “influence” (line 30). For what concern the term “predictor” (lines 27-28), it has not been changed since it is correct with regards to the statistical analysis that has been conducted.

Abstract: “slowly correlated” – non-English; did the authors mean “weakly associated”?

Reply. The term has been changed

Introduction: “complacency .. of vaccination” – unclear.

Reply. According to the “3Cs” model of vaccine hesitancy determinants, developed by  the WHO (see reference n. 15), “vaccine complacency exists where perceived risks of vaccine-preventable diseases are low and vaccination is not deemed a necessary preventive action” and “vaccine confidence is defined as trust in 1) the effectiveness and safety of vaccines; 2) the system that delivers them, including the reliability and competence of the health services and health professionals and 3) the motivations of the policy-makers who decide on the needed vaccines”. The period has been changed in order to clarify the concept, as follow: “high confidence in vaccination programmes, together with low complacency and high convenience of vaccine, are crucial for maintaining high coverage rates” (line 62).

There is no explanation to the following sentence: “From a theoretical point of view, health and vaccine literacy can be considered as determinants of vaccine confidence [26,27]. “ - it is definitely not a determinant, as all the variables are measured in a non-experimental design. The authors in their systematic review in 2018 (ref #26) concluded that the relationships “remain unclear”. How has it become a determinant? Which theory? Unfounded, unfortunately.

Reply. Thanks for the comment. In the 3Cs model, “confidence is defined as trust in 1) the effectiveness and safety of vaccines; 2) the system that delivers them, including the reliability and competence of the health services and health professionals and 3) the motivations of the policy-makers who decide on the needed vaccines”. In one of the newest definition of health literacy, namely that included in the Sørensen Integrated Model, “health literacy is linked to literacy and entails people's knowledge, motivation, and competence to access, understand, appraise, and apply health information in order to make judgments and take decisions in everyday life concerning healthcare, disease prevention, and health promotion to maintain or improve quality of life during the life course”. Moreover, vaccine literacy can be defined as “not simply knowledge about vaccines, but also developing a system with decreased complexity to communicate and offer vaccines as sine qua non of a functioning health system”. So, starting from these definitions, and from the logical and conceptual framework that underlie these definition, health and vaccine literacy could be considered as determinants of vaccine confidence. For this reason, we have stated “from a theoretical point of view”. On the other hand, we know that theoretical and logical frameworks and models have to be validated by field studies, especially by quantitative, reproducible researches that use validated measurement tools. The results of that testing should allow researchers to further refine their original hypothesis, and then retested it. That process can and should be repeated as many times as necessary to arrive at a reasonable degree of certainty. The studies conducted until now have not validated the theoretical model that suggests a relationship between health or vaccine literacy and vaccine confidence because, to the best of our knowledge, no previous studies have measured both vaccine confidence and health or vaccine literacy. In conclusion, in our opinion, at the state of the art there are not sufficient data to confirm or not the theoretical model that underlie the relationship between health/vaccine literacy and vaccine confidence.

Please, consider that the conclusion of our previous systematic review (reference n. 26) was “the relationship between HL and vaccination remains unclear”, since the review aims to describe the state of the art in the relationship between health literacy and the attitude towards vaccines, intention to vaccinate, and vaccine uptake. In fact, no studies measuring vaccine hesitancy and vaccine confidence were included since they were not found. In our opinion, the relationship between health and vaccination remains unclear primarily because, by now, no valid measurement tools of vaccine literacy have been developed.

Methods, under data analysis. There is no information on “uptake” but rather on “reported uptake”. It’s all self-reported. Change that.

Reply. In the methods, results and discussion and conclusion “uptake” has been changed with “reported uptake”, where appropriate (lines 171, 179, 182, 185, 225, table 2, lines 234, 260, 271, table 4, lines 275, 282, 288, 321, 379, 380).

Methods, under data analysis, regression. It would make sense to divide the variables to categories and into 3 steps: demo, background (e.g., having diseases), and perceptual. Not just adjustment for age, sex and language (proxy for immigration?).

Reply. In the multivariate analysis, we have included, as covariates, only the variables with statistically significant association with the outcome variable (reported vaccination uptake, as multinomial). Using this approach, that is widely used, the covariates to be included were: suffering from renal chronic diseases, suffering from autoimmune diseases, living with elderly people, and the VCI score. Moreover, the IMETER adjusted-score was added since it was significantly associated with the VCI score at the univariate and according to the aim of the study. Age and sex are widely used as adjustment variables and mother language was added as adjustment variable since both the questionnaire was submitted in Italian and the IMETER is a word recognition-based test. Regarding the specific question, is the reviewer asking to add in the multinomial (polytomous) logistic regression model all the variables we have analyzed in the univariate analysis?

I do not think that Figure 1 contributes to the manuscript. Why is it presented? There is no “story” in it. Authors may consider removing it.

Reply. Figure 1 reported the graphical representation of the correlation analysis between the IMETER adjusted-score and the VCI, by sub-groups. In our opinion, in line with what has been expressed by the Reviewer n. 1, the figure is useful since it provides a visual, impactful representation of the mutual distribution of the two scores.

Under discussion. Refrain from using the words “affect” “influence”. Inappropriate in a cross-sectional design.

Reply. The terms have been changed according to the suggestion (lines 281-282, 340)

Under discussion. Pls discuss the findings on the association between HL and vaccination in the context of findings on the association between HL and health behaviors. See the reviews by Berman (2011) and Neter & Brainin (2019).

Reply. Unfortunately, we did not find the two suggested reviews. Could the reviewer please add more details about the two references, in order to find them? Nonetheless, the relationship between HL and vaccination in the context of those between HL and health behaviors has been added as a matter of discussion, including also three references (lines 325-328).

Under discussion. “since the results of this study reported that in some cases
high confidence in vaccine had not led to vaccination uptake and vice versa” – unclear. Do the authors refer to the fact that there is distribution?
To the association bot being 1?

Reply. We refer to this result: “Nonetheless, among those who reported high level of VCI score (higher than 75°centile, equal to 2.3), 62 (36.4%) never get vaccinated and among those who reported low level of VCI score (lower than 25°centile, equal to 1.1) only one (0.6%) always get vaccinated and 25 (15.1%) sometimes get vaccinated.” (lines 246-249)

The authors explain the lack of association between HL and confidence in vaccination only in terms of cognitive and behavioral (skill) variables. Pls consider also emotions.

Reply. Thanks for the remark. This aspect has been added in the discussion. Please, note that in this study HL has been measured using an objective measure of functional health literacy. Maybe, the use of a subjective measure of functional, interactive and critical HL should result in an increase of the association between HL and vaccine confidence. This aspect has been added in the discussion (lines 353-361).

/typo: “people.. does” into “people .. do”

Reply. Done (line 341)